# Exploring the Efficacy of Benzimidazolone Derivative as Corrosion Inhibitors for Copper in a 3.5 wt.% NaCl Solution: A Comprehensive Experimental and Theoretical Investigation

**DOI:** 10.3390/molecules28196948

**Published:** 2023-10-06

**Authors:** Mohamed Adardour, Mohammed Lasri, Marouane Ait Lahcen, Mohamed Maatallah, Rachid Idouhli, Mohamed M. Alanazi, Sanae Lahmidi, Abdesselam Abouelfida, Joel T. Mague, Abdesselam Baouid

**Affiliations:** 1Laboratory of Chemistry Molecular, Department of Chemistry, Faculty of Sciences Semlalia, Cadi Ayyad University, B.P. 2390, Marrakech 40001, Morocco; m.aitlahcen.ced@uca.ac.ma (M.A.L.); m.maatallah@uca.ma (M.M.); baouid@uca.ac.ma (A.B.); 2Applied Chemistry and Biomass Laboratory, Department of Chemistry, Faculty of Sciences Semlalia, Cadi Ayyad University, B.P. 2390, Marrakech 40001, Morocco; m.lasri.ced@uca.ac.ma (M.L.); rachid.idouhli@uca.ac.ma (R.I.); abouelfida@uca.ac.ma (A.A.); 3Department of Pharmaceutical Chemistry, College of Pharmacy, King Saud University, P.O. Box 2457, Riyadh 11451, Saudi Arabia; mmalanazi@ksu.edu.sa; 4Laboratory of Heterocyclic Organic Chemistry, Department of Chemistry, Faculty of Sciences, Mohammed V University in Rabat, Rabat 10106, Morocco; lahmidi_sanae@yahoo.fr; 5Department of Chemistry, Tulane University, New Orleans, LA 70118, USA; joelt@tulane.edu

**Keywords:** benzimidazolone, corrosion inhibitor, electrochemical techniques, adsorption

## Abstract

This study focuses on the synthesis, theoretical analysis, and application of the corrosion inhibitor known as benzimidazolone, specifically 1-(cyclohex-1-enyl)-1,3-dihydro-2*H*-benzimiazol-2-one (CHBI). The structure of CHBI was determined by X-ray diffraction (XRD). The inhibitory properties of CHBI were investigated in a 3.5 wt.% NaCl solution on pure copper using various electrochemical techniques such as potentiodynamic polarization curves (PDPs) and electrochemical impedance spectroscopy (EIS), as well as scanning electron microscopy with energy dispersive X-ray spectroscopy (SEM-EDX), UV-visible spectroscopy, and theoretical calculations. The obtained results indicate that CHBI is an excellent inhibitor, exhibiting remarkable effectiveness with an inhibition rate of 86.49% at 10^−3^ M. To further confirm the extent of adsorption of the inhibitory molecule on the copper surface, density functional theory (DFT) and Monte Carlo (MC) simulation studies were conducted. The results of this study demonstrate the synthesis and characterization of CHBI as a corrosion inhibitor. The experimental and theoretical analyses provide valuable insights into the inhibitory performance of CHBI, indicating its strong adsorption on the copper surface.

## 1. Introduction

Organic inhibitors are an exceptionally effective and cost-efficient approach in the wide range of methods employed for corrosion prevention. This approach has gained widespread application for mitigating copper corrosion in multiple fields. Reviewing the existing literature unveils that the existence of heterocyclic compounds acts as active adsorption sites for facilitating the interaction between the corrosion inhibitor and copper through the exchange of electrons. In addition, heterocyclic compounds within aromatic structures exhibit effective corrosion inhibitory properties in a variety of corrosive environments [1,2,3,4].

Benzimidazolone is a heterocyclic aromatic organic compound consisting of the fusion of benzene and imidazolone rings. This heterocyclic framework serves as the foundational structure for a significant class of compounds, showcasing intriguing biological activities across a wide range of therapeutic domains. Recent research has found that benzimidazolone derivatives exhibit anti-HBV activities [5,6] as well as anticonvulsant [7] and antimicrobial [8] activities. Additionally, benzimidazole derivatives also play an important role in cancer treatment [9,10]. Benzimidazole derivatives featuring heteroatoms such as sulfur, nitrogen, and oxygen have emerged as some of the most widely employed corrosion inhibitors for various aggressive environments, particularly as effective coatings [11,12,13]. Benzimidazolone derivatives have a wide range of potential applications, including pharmaceuticals, inhibitors, herbicides, pigments, and fine chemicals [14]. In addition, benzimidazoles, are also of interest in advanced materials science, such as nonlinear optics [15]. Furthermore, due to the structure of benzimidazole, it can form hydrogen bonds with enzymes and biological receptors, take participate in π–π and hydrophobic interactions, and serve as a ligand for various metal ions.

The literature abounds with examples that underscore the versatility and effectiveness of benzimidazole and its derivatives as corrosion inhibitors. Their unique molecular structure and compatibility with various industries and environments have solidified their role as essential components in ongoing efforts to protect valuable assets and infrastructure from the ravages of corrosion [16,17,18,19]. The use of benzimidazole compounds as corrosion inhibitors for copper at different concentrations of sodium chloride has been reported [20,21]. Recent experiments on corrosion [22,23,24,25] have shown that benzimidazole derivatives possess corrosion-inhibiting properties (Figure 1). According to the literature, most organic compounds are effective corrosion inhibitors, which are widely used to protect copper alloys against corrosion in aqueous environments [22,23,24,25,26,27,28]. Analogous compounds were employed as corrosion inhibitors for copper in a 3.5 wt.% NaCl solution at a temperature of 25 °C. The results suggest that introducing a nitrogen atom into the imidazole ring enhances the inhibitory effect, which is mainly due to the heightened adsorption energy. The three investigated inhibitors demonstrated a propensity to adsorb onto the copper surface in a parallel fashion, leading to optimal inhibition effects. This study holds the potential to introduce novel inhibitors with superior inhibitory performance and enhanced effects. Appa Rao et al. [29] documented the corrosion protection achieved by self-assembled monolayers of benzimidazole derivatives on copper surfaces immersed in a NaCl solution. Pareek et al. [30] investigated benzimidazole analogs as a copper corrosion inhibitor in 3.5 wt.% NaCl solution, using both experimental and theoretical methods. This compound has a corrosion inhibition efficiency of 92.79% at 0.80 mM.

Benzimidazole and its derivatives have also been employed as corrosion inhibitors for mild steel in an environment containing 1 M HCl [31,32]. These heterocyclic compounds are effective corrosion inhibitors for metals in various corrosive environments. In this context, the study carried out by Chaouiki [33] demonstrated the inhibition potential of certain benzimidazole compounds for mild steel in a HCl solution. The results showed that both benzimidazole derivatives acted as good corrosion inhibitors. Recently, Ech-chihbi et al. [34] conducted a study in which they examined organic compounds incorporating the benzimidazole ring. These heterocyclic compounds were assessed for their corrosion inhibition potential on mild steel in a 1 M HCl solution, utilizing both experimental and computational techniques. Within this context, the findings revealed that these compounds exhibited a substantial inhibition efficiency ($\eta_{EIS}$%) ranging from 90.4% to 95.7% at $10^{-4}$ M.

In the present study, the adsorption and inhibitive effects of CHBI on the corrosion of copper in a 3.5 wt.% NaCl solution were investigated. The corrosion inhibition effectiveness and adsorption characteristics of CHBI were examined through a combination of experimental methods, including Potentiodynamic Polarization analysis (PDP), Electrochemical Impedance Spectroscopy (EIS), and a morphological study using SEM-EDX. Additionally, a theoretical approach was employed, utilizing Density Functional Theory (DFT) calculations at the B3LYP/6–311G(d, p) level of theory, along with molecular dynamics simulation methods. These comprehensive methods provided a deeper understanding of CHBI’s corrosion inhibiting capabilities and its interactions with the copper surface.

## 2. Result and Discussion

### 2.1. Chemicals and Synthesis Procedure

The synthesis procedure for CHBI is illustrated in Figure 1. It includes the condensation of o-phenylenediamine (**1**) and keto-ester (**2**) in refluxing xylene. The structure of CHBI was determined by analytical and spectroscopic analyses. The known CHBI was identified based on the spectroscopic data reported in the literature [35,36]. The structure of CHBI was confirmed by XRD measurements on single crystals as previously reported in another study [37].

### 2.2. X-ray Structure Description and Optimized Geometry

Specific information regarding the X-ray crystal data, structure, and refinement of CHBI is presented in Table 1. Appendix A related to the compound are available and have been deposited with the Cambridge Crystallographic Data Center (CCDC) under the accession number **2280269**. Suitable crystals were grown by a slow evaporation of an ethanolic and dichloromethane (2:1, *v*/*v*) solution of the compound. The benzimidazolone moiety in CHBI is planar to within 0.0136(7) Å with N2 above the distance from the mean plane. A crystallographic analysis revealed that the cyclohexenyl ring in the compound is disordered and adopts two opposite conformations in a ratio of 0.5154(17)/0.4846(17). To account for this disorder, both components were refined with restraints applied to ensure that their geometries remained comparable during the refinement process. These restraints help to accurately represent the disorder in the cyclohexenyl ring and provide a comprehensive description of the molecule’s structure.

A perspective view of CHBI is shown in Figure 2. In the crystal, paired N2–H2A···O1 hydrogen bonds (Table 2) form inversion dimers that are connected by C4–H4···Cg2 interactions (Table 2) and slipped π-stacking interactions between C1···C6 and C1/C6/N1/C7/N2 rings across inversion centers resulting in a layered structure, as shown in Figure 3.

### 2.3. Electrochemical Measurement

#### 2.3.1. Potentiodynamic Polarization

Polarization curves for copper in a corrosive aqueous NaCl solution (3.5 wt.%) without and with different doses of CHBI are shown in Figure 4a, and the electrochemical parameters adjusted by Tafel analysis are given in Table 3, which include $E_{corr}$ (corrosion potential), $i_{corr}$ (corrosion current density), βa (anodic Tafel slope), βc (cathodic Tafel slope), and $n_{1}$% (corrosion inhibition efficiency). The corrosion inhibition efficiency was determined from the polarization results and can be used to calculate the corrosion inhibition capacity according to Equation (1), in which $i_{corr}^{{}^{\circ}}$ and $i_{corr}$ represent the current densities without and with the inhibitor, respectively.
(1)$$n_{1}\;\%=(1-\frac{i_{corr}}{{i{}^{\circ}}_{corr}})\times 100$$

In the absence of an inhibitor, a higher current density value is observed, which is explained by the anodic oxidation of Cu to ${Cu}^{+}$, which forms a soluble complex ${CuCl}_{2}^{-}$ with ${Cl}^{-}$ ions as shown in the following equations (Equations (2)–(4)) [38,39,40,41]:(2)$$Cu\to {Cu}^{+}+e^{-}$$
(3)$${Cu}^{+}+{Cl}^{-}\to CuCl\;insoluble\;complex$$
(4)$$CuCl+{Cl}^{-}\to {CuCl}_{2}^{-}\;soluble\;complex$$

On the cathodic side, we observe the irreversible oxygen reduction controlled by charge transfer as shown below (Equation (5)) [42,43].
(5)$$O_{2}+2H_{2}O+{4e}^{-}\to {4OH}^{-}$$

As depicted in Figure 4a, the corrosion potential ($E_{corr}$) of the copper electrode changed negatively when the inhibitor CHBI was added, compared to that of the blank solution. Furthermore, an interesting observation is that the cathodic branches of the polarization curves remained parallel to the blank solution, suggesting minimal alterations in the cathodic reaction. However, the anodic branches displayed more pronounced variations when the inhibitor was introduced at different concentrations. This observation suggests that CHBI selectively affects the anodic corrosion reactions, highlighting its potential as a corrosion inhibitor for copper under the examined conditions. Based on the data presented in Table 3, with rising inhibitor concentration, the corrosion inhibition efficiency ($\eta_{1}\%$) shows an upward trend. The largest inhibition efficiency observed was 86.47% at 1 mM/L, indicating that a higher inhibitor concentration results in greater corrosion protection for copper surfaces. This suggests that the inhibitor can rapidly attach to the active sites on the copper surface during the 1-h immersion period, resulting in the formation of an anchored barrier film. This film effectively delays the redox reaction and exerts an active protective effect on the copper surface, contributing to the enhanced corrosion inhibition observed at higher inhibitor concentrations [44,45].

#### 2.3.2. Electrochemical Impedance Spectroscopy

The EIS was utilized to analyze the adsorption, protective effect, and kinetics of the corrosion process facilitated by CHBI on the copper surface. To investigate the effect of the inhibitor concentration on the inhibition efficiency, the inhibitor concentration was varied from 10^−3^ to 10^−5^ mol/L. The efficiency of the corrosion protection efficiency was determined from the results of the measurements of EIS ($n_{2}$%), which can be used to calculate the corrosion protection ability according to Equation (6), in which $R_{p}^{{}^{\circ}}$ and $R_{p}$ indicate the polarization resistances without and with inhibitor, respectively.
(6)$$n_{2}\;\%=(1-\frac{{R{}^{\circ}}_{p}}{R_{p}})\times 100$$

In Figure 4b, the Nyquist diagrams illustrate the impedance behavior of CHBI. The charge transfer resistance ($R_{ct}$) observed in the high-frequency loop of the Nyquist diagrams characterizes the mass transfer from the electrode surface, which occurs due to diffusion [46,47]. Meanwhile, the Warburg impedance (W) observed in the low-frequency region signifies the mass transfer from the electrode surface, which is also attributed to diffusion processes [48,49]. These EIS measurements provide valuable insights into the interactions between CHBI and the copper surface, shedding light on the inhibition efficiency of different inhibitor concentrations. The EIS data were analyzed using an equivalent circuit model, as illustrated in Figure 4d, and the corresponding parameters were calculated, as presented in Table 4. The circuit model comprises two Constant Phase Elements (CPE) connected in series with the uncompensated resistance $R_{s}$. This combination represents both the solution resistance and the resistance attributed to the electrical connections within the system. By utilizing this circuit model, the EIS analysis provides valuable information about the behavior of CHBI on the copper surface, offering insights into the inhibitive action and kinetics of the corrosion process at different inhibitor concentrations [50]. $R_{ct}$ represents the charge transfer resistance of copper dissolution between the metal surface and the external Helmholtz plane at the metal–solution interface. The Nyquist plots show an apparent deviation from the ideal semicircle, which is attributable to surface inhomogeneity due to surface roughness, impurities, and inhibitor adsorption [48].

In EIS, a CPE is frequently employed in place of an ideal capacitor to accommodate the frequency dispersion response [51]. In this context, the ${CPE}_{dl}$ is utilized to model the double-layer capacitance at the metal–solution interface. Additionally, the ${CPE}_{f}$ denotes the CPE associated with the surface film, where $R_{f}$ represents the corresponding film resistance. Moreover, the Warburg element (W) is employed to describe the diffusion mass transfer from the metal surface. By incorporating these elements in the equivalent circuit model, EIS provides a more accurate representation of the frequency-dependent behavior and facilitates a comprehensive analysis of corrosion inhibition and surface film formation on the copper electrode.

In Figure 4c, the phase angle versus log f (frequency) curves demonstrate a gradual increase in phase angle values, which approach approximately −75°. This behavior is typical of a non-ideal capacitor [52]. In Figure 4a,b, the curves of log |Z| (modulus of impedance) versus log f curves exhibit three distinct sections: (i) the high-frequency region, where the log |Z| values are smaller and remain nearly constant while the phase angle approaches zero. (ii) The medium-frequency region, which exhibits a linear relationship between log |Z| and log f values, and the phase angle reaches its maximum value. (iii) The low-frequency region, in which the log |Z| values become independent of log f, indicating the presence of Warburg impedance. This points toward a diffusion-controlled reactant transfer from the bulk electrolyte to the metal/electrolyte interface or transfer of soluble species from the interface to the bulk solution. Notably, the significantly higher phase-angle values observed in the presence of CHBI, compared to its absence, reflect its inhibitory performance. This suggests that CHBI effectively impedes the corrosion process, resulting in a noticeable alteration in the impedance behavior. This alteration can be attributed to the formation of a protective barrier on the surface of the copper.

#### 2.3.3. Adsorption Isotherm

The ability of organic corrosion inhibitors to adhere to metal surfaces is one factor that affects how much corrosion is reduced. Adsorption isotherms provide much essential information [53]. Like the adsorption isotherms (Langmuir), in this section we used the Langmuir adsorption isotherm using the corrosion inhibition efficiency values provided by the polarization potentiodynamic and EIS [54]. The Langmuir adsorption isotherm is a straight-line plot of C/θ with respect to the corrosion inhibitor concentration, as shown in Figure 5. The adsorption constant values ($K_{ads}$) of the inhibitor used were estimated according to the following equation (Equation (7)) and are given in Table 5.
(7)$$\frac{C_{inh}}{ɳ}=\frac{1}{K_{ads}}+C_{inh}$$
where ɳ is the surface coverage, *C* is the inhibitor concentration, and $K_{ads}$ is the adsorption equilibrium constant.

The Langmuir isotherm plots of CHBI are shown in Figure 5. The plots of C/ɳ as a function of *C* for pure copper in a 3.5wt.% solution at 298 K show a straight line. The linear correlation coefficient with *R*^2^ > 0.999 confirms that the Langmuir isotherm adequately accommodates the experimental data. The values of $K_{ads}$ are used to calculate the values of the change in adsorption standard free energy (${\Delta G}_{ads}^{{}^{\circ}}$) using Equation (8) [55]:(8)$${\Delta G}_{ads}^{{}^{\circ}}=-RTLn(K_{ads})$$
where *R* is the universal gas constant and *T* is the absolute temperature (K).

Table 5 shows the values of $K_{ads}$ and ${\Delta G}_{ads}$ calculated from the EIS and PDP measurements. The adsorption process is more spontaneous when ${\Delta G}_{ads}$ is negative [56]. ${\Delta G}_{ads}$ values of 20 kJ/mol or less are typically associated with physisorption, while values of 40 kJ/mol or more are associated with chemisorption [57]. The values of ${\Delta G}_{ads}$ in this study indicate both chemisorption and physisorption, with a predominance of chemisorption.

#### 2.3.4. Temperature Influence

Electrochemical investigations were conducted to explore the effect of temperature on the corrosive degradation of pure copper when immersed in a 3.5 wt.% NaCl solution containing an inhibitor. The temperature range studied was between 298 and 323 K, employing PDP methods, as shown in Figure 6a. As shown by the outcomes in Table 6, both the corrosion current density values, with and without the inhibitor, exhibit an increase as the temperature rises. This rise in temperature leads to a shift in the inhibitor adsorption equilibrium, favoring a desorption process and consequently leading to a reduction in surface coverage [58,59]. As a result, the inhibitor surface conformation order decreases and the corrosive ion mobility increases [60]. At 323 K, the CHBI has a significant corrosion inhibition efficiency (38.88%) when compared to a blank, as shown in Table 6.

The inhibitor protection mechanism is based on thermodynamic activation characteristics [61]. The Arrhenius Equation (9) was used to calculate the corrosion activation energy ($E_{a}$) in 3.5 wt.% NaCl with and without CHBI:(9)$$i_{corr}=Ae^{-\frac{Ea}{RT}}$$
where *T*, *A*, and *R* denote the absolute temperature, the Arrhenius factor, and the universal gas constant, respectively.

The CHBI activation energy ($E_{a}$) increased from 11.00 kJ.mol^−1^ to 34.95 kJ.mol^−1^ as measured by the Arrhenius diagrams in Figure 6b. The energy barrier for the corrosion reaction was increased, which improved corrosion inhibition [62]. A decrease in inhibition efficiency with rising temperature and a rise in $E_{a}$ when the inhibitor is present serve as evidence that a physical (electrostatic) adsorption film was formed (Table 7).

#### 2.3.5. Morphological Study by SEM-EDS

A polished circular section of copper, measuring 0.5 cm^2^ in area, was submerged for a period of 24 h at a temperature of 298 K in a 3.5 wt.% NaCl solution. This process was carried out both without and with the addition of 1mM of inhibitor (CHBI). After immersion, the surface homogeneity of the copper was meticulously inspected under both conditions to assess the extent of the reduction in NaCl activity in the presence of the inhibitor. The samples were analyzed by SEM for surface examination after being washed with doubly distilled water and dried. Furthermore, the atomic percentage of the elements was investigated using an EDX analyzer due to the inhibitor deposited on the copper surface [30].

To comprehend the inhibitory effect, it is crucial that the organic molecule forms a strong bond with the metal substrate, resulting in the formation of a protective layer on the metal surface [63]. SEM and EDX experiments were performed to see if the CHBI molecule actually gets adsorbed onto the copper surface. By examining and determining if CHBI is present on the copper surface, these analytical approaches can shed important light on how the inhibitor interacts with the metal.

Figure 7a illustrates the copper surface after immersion in a 3.5 wt.% NaCl solution, revealing flaws such as pits, pores, and fractures caused by the dissolution of copper. The SEM micrograph of the surface immersed in NaCl for 24 h with 10^−3^ M of CHBI, which is shown in Figure 7b, exhibits the formation of a thin protective layer due to the adsorption of CHBI on the copper surface. According to our results, the presence of CHBI reduced the damage caused by cracking and pitting on the copper surface.

In the EDX spectrum (Figure 7c) of unprotected copper in the presence of the corrosive medium, we notice a peak for copper and another for oxygen, as well as other peaks that can be attributed to the generation of chlorides and sodium. Figure 7d shows the EDX spectrum of the surface in the presence of CHBI, and we observe peaks corresponding to carbon, nitrogen, and oxygen, which is explained by the formation of a protective film on the surface of the copper.

### 2.4. Results of Density Functional Theory and Molecular Dynamics Simulations

#### 2.4.1. Analysis of Conceptual DFT Indices

The structure of CHBI in its most stable configuration was determined by optimization at the 6-311G(d,p) level (Figure 8a). The calculated free enthalpy (−432,062.71577 Kcal/mol) indicates the thermodynamic stability of the molecule. After optimization, different quantum chemical descriptors characterizing the reactivity of CHBI were calculated using the above equations (Table 8).

The data in Table 8 indicate that CHBI has a high reactivity and a consistently good performance as an inhibitor. Considering the difference in electronegativity, CHBI easily interacts with the metal surface and can resist change or deformation in it since it exhibits high global hardness (2.662 eV) and low global softness (0.376 eV) values.

The interaction between CHBI and the surface is facilitated by a donation of electrons, which takes place from CHBI to the metal (∆N = 0.74 > 0). This result is supported by the energies of the HOMO (−5.736 eV) and the LUMO (−0.412 eV), which when compared with those of Cu [63] indicate a substantial interaction between the HOMO of CHBI and the LUMO of the metal, indicating that there is an electron transfer from CHBI to the empty orbitals of the metal.

The electron density distribution of the HOMO and the LUMO of CHBI, shown in Figure 8b, is located essentially on the CHBI ring including the N and O heteroatoms. Therefore, these atoms represent the most reactive centers of the CHBI and participate in its interaction with the metallic surface via their lone pairs. The interaction is further strengthened by the π electrons of the phenyl ring, and the values of w^−^ (3.64 eV), w^+^ (−1.96 eV), and ∆w ± (1.7 eV) also support the ability of CHBI to donate electrons to the metal surface.

#### 2.4.2. MEP Surface

The MEP is a very helpful tool for understanding how an inhibitor can interact with a metal surface by revealing its nucleophilic and electrophilic centers. The MEP is characterized by a colored zone according to the electrostatic potential. The maximum negative region (red areas) indicates electron-rich (nucleophilic) sites, while the maximum positive region (blue areas) characterizes electron-deficient (electrophilic) sites. Figure 8c shows the MEP of CHBI in its optimized form, through which the carbonyl oxygen (O1) and the phenyl ring (rich in π electron) are identified as electron-rich centers. These can interact with the metal surface as nucleophiles contributing electrons to its vacant sites. Similarly, the carbon and hydrogen atoms of the cyclohexenyl ring, where the blue zone is located, are likely to interact with the surface as electrophilic centers, accepting electrons from the electron-rich sites on the surface.

#### 2.4.3. Condensed Fukui Functions

The active sites and the local reactivity of the inhibitor molecule can also be understood using other descriptors such as the Fukui functions (f^+^, f^−^) where the electrophilic/nucleophilic character can be described as a function of the sign of the Fukui function. The values of the Fukui f^+^ and f^−^ functions are listed in Table 9. According to the table, the electrophilic attack site of CHBI was identified based on the positive charge value of the most reactive sites capable of accepting electrons during metal–inhibitor interactions. Furthermore, the CHBI Fukui function f^−^ illustrated the reactivity of the different areas within CHBI concerning the electrophilic reaction with electrophiles. A higher value of f^−^ suggested a more reactive site in those regions, indicating a greater tendency for electrophilic attack and implying that the atom in that region is more likely to donate electrons. This shows that the atoms of C6 and C12 exhibit the maximum values of f^+^ (0.122 and 0.133, respectively), indicating that these atoms are most likely to admit electrons from the copper surface to form feedback bonds, thus enhancing the interaction between the metal surface and the inhibitor. The atoms O1, N2, N3, and C8 have the largest values of f^−^ (0.134, 0.022, 0.029, and 0.069, respectively), making these atoms the most nucleophilic and effective adsorption sites. These results indicate the ability of the inhibitor CHBI to provide electrons throughout the molecule and thus explain the significant inhibitory effect of this type of organic inhibitor in aqueous solution.

### 2.5. In Silico Approaches for Environmental Toxicity

The toxicity characteristic of the CHBI examined was assessed using log (IGC_50_), which indicates the concentration required to inhibit 50% of CHBI growth in aquatic species. The results of the toxicity test are provided in Table 10. When examining Table 10, it becomes apparent that the logarithmic value (IGC_50_) obtained for CHBI was smaller than the study’s maximum concentration (10^−3^ M). This result suggests that the compounds under investigation showed evidence of having corrosion-inhibiting abilities that are friendly to the environment.

### 2.6. Molecular Dynamics Results

The inhibitory effect of the CHBI near the Cu surface was theoretically investigated by Monte Carlo (MC) and MD simulations. This simulation was carried out to analyze the behavior of CHBI in its neutral and protonated forms on the surface of Cu(110) in vacuum and in the presence of the corrosive medium $\left\langle H_{2}O|{Na}^{+}|{Cl}^{-}\right\rangle$.

The most stable equilibrium adsorption configurations of the neutral and protonated forms of CHBI on the Cu(110) surface are shown in Figure 9, and the various terms of the associated energy components are listed in Table 11. The data depicted in Figure 9 indicate that CHBI exhibits facile adsorption on the Cu(110) surface, adopting a parallel orientation. Such a position allows optimal interactions between heteroatoms and π-electrons with the metal surface, increasing surface coverage when interacting with the copper surface. This adsorption becomes stronger when the molecule is protonated (−77.08850781 and −98.97803808 Kcal/mol for the neutral and protonated forms, respectively), which is justified by the calculated energy values (Table 11). The parallel configuration of the molecule is also conserved in the presence of the corrosive medium. Based on the calculated energies listed in Table 12, the adsorption becomes stronger: −2255.320 and −2405.338 Kcal/mol for the two forms, respectively. These results are confirmed by the interaction and binding energies calculated under the equilibrium condition in an aqueous medium. The interaction and binding energies between the surface and the inhibitor are determined as follows (Equations (10) and (11)):(10)$$E_{interaction}=E_{total}-E_{surface+solution}-E_{inhibitor}$$
(11)$$\begin{array}{cc} E_{interaction}= & {-E}_{Binding} \end{array}$$
where $E_{total}$ is the total energy of the entire system under study, $E_{surface+solution}$ represents the energy of the Cu(110) surface and solution, and $E_{inhibitor}$ is related to the energy of the inhibitor.

Table 13 presents the calculated $E_{interaction}$ and $E_{binding}$ values. The presence of negative $E_{interaction}$ values and positive $E_{binding}$ values indicates the spontaneous and robust adsorption process of both CHBI and protonated CHBI. An alteration in the trend of diminishing interaction and adsorption is noticed on the surface. Notably, higher values correspond to a better inhibition performance of the molecule. Therefore, from the calculated values, it can be concluded that our inhibitor readily adsorbs on the copper surface.

In the most stable equilibrium configurations, the aromatic ring of CHBI was aligned parallel to the Cu surface. Moreover, the oxygen and nitrogen atoms were oriented toward the metal surface as shown in Figure 9. This flat orientation is possibly due to the formation of coordination and back bonding between the Cu metal and CHBI inhibitor. It is herein evident that unoccupied copper orbitals (3d) will prefer to accept electrons from the adsorbed CHBI. In addition, CHBI has a lone pair of electrons on the active centers (NH, N, O), as well as π-electrons in the benzene ring. These electrons clouds provide sufficient electronic charges to the vacant orbitals (3d) of Cu and form stable coordination bonds (chemisorption). The spontaneous position of the CHBI inhibitor allows for maximum interactions between its active sites (the heteroatoms and the π electrons of the phenyl ring) and the metal surface. As a result, this increases its coverage and promotes the formation of a protective layer on the copper surface.

The formation of this monolayer suggests a chemisorption process between the inhibitor CHBI and the copper surface, which is consistent with the radial distribution function (RDF) results. The RDF was used to calculate the bond length between the atoms of CHBI and those on the Cu surface and to determine the type of bonds formed. Figure 10 shows the variations of the RDF as a function of bond length (r). The link length values varied from 2.68 to 2.98 Å, which indicates that the interactions developed are of the chemisorption type.

## 3. Materials and Methods

### 3.1. Chemistry

The synthesis of CHBI was successfully accomplished following the steps outlined in Figure 1. All the chemical reagents utilized in the synthesis were obtained from Sigma Aldrich (St. Somaprol, Morocco) and were used without any further purification. The purity of CHBI inhibitor was determined by thin-layer chromatography (TLC) and melting point (m.p). IR spectra were recorded on a Bruker Vertex 70 spectrometer as potassium bromide discs. Melting points were taken in an open capillary tube on a Buchi 510 apparatus and were uncorrected. The spectra were recorded with the following instruments: ^1^H NMR (AC-300) and ^13^C NMR (AC-75) spectra were recorded on Bruker spectrometers (Bruker, France) with chemical shift values (d) given in part per million (ppm) relative to TMS (0.00 ppm). Mass spectra: Jeol JMS DX 300. TMS was used as an internal reference. Elemental analysis was performed on a EuroEA Elemental Analyzer.

#### Synthesis of CHBI

The CHBI was prepared according to a reported procedure [37]. A mixture of o-phenylenediamine **1** (23 mmol) and ketoester **2** (26 mmol) in absolute xylene. The reaction was carried out at a reflux temperature for a duration of 5 h (Figure 1). The reaction mixture was then cooled, and the precipitated solid was filtered and recrystallized from ethanol to obtain CHBI. The CHBI was obtained in the form of a white solid. Yield: 92%. m.p: 181–183 °C (ethanol). IR (KBr, ν (cm^−1^), 3413 (imidazole-NH), 1602 (C=O). ^1^H NMR (300 MHz in CDCl_3_) (δ ppm): 1.69, 1.78, 2.22, 2.33 (m, 8H, 4CH_2_), 5.90 (m, 1H, =CH), 6.89–7.17 (m, 4H, H-Ar), 10.77 (s, 1H, NH). ^13^C NMR (75 MHz in CDCl_3_) (δ ppm): 21.65, 22.60, 24.76, 26.87 (4CH_2_), 127.76 (1C, =CH), 108.71, 110.09, 121.38, 121.69 (CH-Ar), 128.44, 130.54, 132.07 (=C), and 154.06 (C=O). HRMS of [M+H]+ *m*/*z* 215.1179, calcd for C_13_H_14_N_2_O, 215.1140. Elem. Anal. Calcd for C_13_H_14_N_2_O: C 72.87%, H 6.59%, N 13.07%. Found: C 71.63%, H 6.53%, N 13.54%.

### 3.2. X-ray Analysis

The collection of single-crystal X-ray data was performed using a Bruker D8 VENTURE PHOTON 3 CPAD diffractometer with Cu-Kα radiation (λ = 1.54178 Å) at a temperature of 150 K. The intensity data obtained were converted to F^2^ values using SAINT [64]. Subsequently, a multi-scan absorption correction was applied, and the equivalent reflections were merged using SADABS [65]. The structure was initially solved using dual-space methods with SHELXT [66]. Afterward, full-matrix, least-squares procedures were employed to refine the structure using SHELXL [67]. Hydrogen atoms bonded to carbon were incorporated as riding contributions in idealized positions, with their isotropic displacement parameters linked to those of the attached atoms. This approach is commonly used in crystallographic refinement to simplify the process and improve accuracy. The hydrogen atom attached to nitrogen was initially identified in a difference map and further refined using the DFIX 0.91 0.01 instruction. The specifics of the X-ray crystal data, the structural solution, and the refinement process are all presented in Table 1. This table likely contains the relevant details necessary for understanding the crystallographic analysis and the determined structure.

### 3.3. Corrosion Methods

#### 3.3.1. Preparation of Materials

In this experiment, a highly pure copper electrode (with 99.9% purity) was immersed in a 3.5 wt.% NaCl solution, which served as a highly corrosive environment. To protect the copper from corrosion, CHBI was used as a corrosion inhibitor at various concentrations: 1, 0.1, 0.05, and 0.01 mM. Before conducting the tests, CHBI was introduced into a blank solution. Prior to the corrosion testing, the copper samples underwent a step-by-step polishing process using different grades of emery paper (ranging from 400 to 2500 grains). The samples were then rinsed with distilled water and anhydrous ethanol and left to dry at room temperature. The assessment of CHBI’s corrosion inhibition effectiveness was conducted by immersing the copper samples in the corrosive solution for a duration of 1 h. This allowed the researchers to evaluate the inhibitory performance of CHBI under the specified conditions.

#### 3.3.2. Electrochemical Measurements

The electrochemical assessments were conducted using the potentiostat/galvanic PGZ 100 electrochemical workstation, employing the conventional three-electrode system [68]. For this setup, a pure copper electrode with an effective exposed area of 0.5 cm^2^ was utilized as the working electrode. Additionally, a saturated calomel electrode (SCE) was employed as the reference electrode, and a platinum electrode with a surface area of 1 cm^2^ was used as the counter electrode. A 3.5 wt.% NaCl solution was applied to the working electrode for 1 h at 298 K. Following the immersion, two electrochemical tests were conducted: EIS and PDP measurements. For the PDP measurement, the polarization curve was scanned from −1000 mV to +600 mV using a scan rate of 1 mV s^−1^. On the other hand, the EIS was performed across a frequency range spanning from 0.01 Hz to 100 kHz, with an amplitude of 10 mV [69,70]. The EC-Lab software (V10.44) package was utilized to analyze the electrochemical data, employing equivalent electrical circuits and associated fitting parameters. To ensure the reliability of the results, a minimum of three repeated measurements were carried out, aiming to attain good reproducibility.

### 3.4. Theoretical Study Computational Details

#### DFT Study

To understand the performance of CHBI from an electronic point of view, structural optimization was performed, and the electronic properties of CHBI were obtained using DFT implemented in Gaussian 09 [71] with the hybrid B3LYP functional and the 6–311G(d, p) basis set [72,73,74]. The optimization was followed by a frequency calculation to ensure that the optimized structure is at the minimum on the potential energy surface.

The frontier molecular orbitals (FOM), Highest Occupied Molecular Orbital (HOMO), and Lowest Unoccupied Molecular Orbital (LUMO) play a crucial role in the analysis of the chemical reactivity of organic molecules. They are also employed to investigate the inhibitory properties of molecules, as their adsorption is associated with the potential interaction between the HOMO and LUMO of the inhibitory molecule and the metal surface. The mode of interaction can be determined by a transfer of electrons, either by a donation of electrons from the molecule toward the metal, or by a retro-donation of electrons from the metal toward the inhibitor. For this purpose, based on the energies of the frontier orbitals, the reactivity of the inhibitor was studied from quantum chemical parameters such as the band gap (∆Eg), the ionization potential (I), the electron affinity (A), the hardness (η), the softness (σ), and the electronegativity (χ). The electron transfer fraction (∆N) and the electrophilicity index (ω) were also evaluated since these are two characteristics responsible for the possible electronic exchange between the molecules and the surface. The Fukui parameters serve as valuable descriptors that can be utilized to identify the reactive centers involved in the inhibition process. These descriptors cited above were calculated based on the following expressions (Equations (12)–(18)) [75,76,77]:(12)$$∆E_{gap}=E_{LUMO}-E_{HOMO}$$
(13)$$µ=\frac{I-A}{2}\;\mathrm{and}\;\chi =-\frac{1}{2}\left(E_{HOMO}+E_{LUMO}\right)$$
(14)$$\eta =\frac{I-A}{2}\;\mathrm{and}\;\sigma =\frac{1}{\eta }$$
(15)$$I=-E_{HOMO}\;{\mathrm{and}}_{\;}A=-E_{LUMO}$$
(16)$$N=E_{HOMO}-E_{HOMO(TCE)}\;\mathrm{and}\;\omega ={\frac{\mu }{2\eta }}^{2}$$
(17)$$f^{+}=q_{k}\left(N+1\right)-q_{k}\left(N\right)\;For\;nucleophilic\;attack$$
(18)$$f^{-}=q_{k}\left(N\right)-q_{k}\left(N+1\right)\;For\;electrophilic\;attack$$

### 3.5. Toxicity Approach

In silico research was performed using a chemical database and modeling environment website called OCHEM to assess the toxicity of the CHBI molecule. Two subsystems make up this website: a main database of experimental techniques, and a modeling framework [78,79].

### 3.6. Molecular Dynamic Simulation Method

A molecular dynamics (MD) simulation was performed between the inhibitor molecule and the Cu(110) surface using a slab of 5 Å. Simulation boxes of dimensions 25.50 × 25.50 × 38.33 Å^3^ were created, and a vacuum plate of 30 thickness was introduced. The MD simulations were carried out at a constant temperature of 300 K using the Andersen thermostat, a time step of 1 fs, an NVT set, and a simulation time of 2000 ps to reach the equilibrium configuration. Energy minimization and MD calculations in the condensed phase were performed using the Condensed-phase Optimized Molecular Potential for Atomistic Simulation Studies (COMPASS) force field.

## 4. Conclusions

In this study, we synthesized the compound CHBI for copper corrosion inhibition. We found that CHBI is an excellent inhibitor of copper corrosion in a 3.5 wt.% NaCl solution. The inhibition efficiency increases with inhibitor concentration and reaches a maximum of 86.49% at $10^{-3}$ M. The adsorption of CHBI on the copper surface follows the Langmuir adsorption isotherm. Increasing the temperature from 298 to 323 K decreases the inhibition efficiency of CHBI to 38%. MEB-EDX analysis revealed that the adsorption of CHBI on the copper surface forms a protective layer. DFT numerical simulation results showed that CHBI has a strong ability to donate electrons to the copper surface, which confirms its excellent anticorrosive activity. A strong correlation was observed between quantum chemical parameters and the experimentally obtained inhibition efficiencies of CHBI.

These findings not only expand our understanding of CHBI’s role as a corrosion inhibitor but also highlight potential avenues for further research in corrosion protection and inhibitor optimization, holding promise for practical applications in protecting copper and related materials from corrosion.

## Data Availability

Not applicable.

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
