# Peer review of "Exploring the Efficacy of Benzimidazolone Derivative as Corrosion Inhibitors for Copper in a 3.5 wt.% NaCl Solution: A Comprehensive Experimental and Theoretical Investigation"

_molecules, 2023, doi:10.3390/molecules28196948_

Round 1

Reviewer 1 Report

The authors present an interesting study. However the paper is premature for publication, and the reviewer suggested Major revision.

Abstract

"The results obtained suggest that CHBI act as excellent inhibitor, displaying remarkable effectiveness with an inhibition rate of 86.49% at 10−3 M"

The reviewer will not agree that ~85% inhibition efficiency (not inhibition rate) is not remarkable. Please compare with other common inhibitors.

In the Introduction, the authors should focus on copper, not on mild steel. Please rewrite.

The authors used “a saturated calomel electrode (SCE)”, but on all the figures and the text appears ECS?  

Table 5

The Kads highly differ, so how (delta)Gads can be practically the same? Also sign for “kilo” is k, not K.

Tables 3 and 6.

Reviewer: What is the meaning of such a high cathodic Tafel slope (some are positive?) For the diffusion-controlled oxygen reduction reaction Tafel slope is meaningless. How the authors determined corrosion current density???? As authors stated, row 129, “Polarization curves for copper in a corrosive aqueous NaCl solution (3.5 wt.%) with-129 out and with different doses of CHBI are shown in Fig. 3(a), and the electrochemical parameters adjusted by Tafel analysis are given in Table 3,.....” What is the Tafel analysis, it can be applied only to the anodic part of the curve.

Reviewer: When using log functions the correct is to write log (i / mA/cm2) or log (Z / Ohm cm2) not log i (mA/cm2)

Tables, 4 and Fig. 3 Units for Z and R are Ohm cm2 not Ohm/cm2

row 504 “To ensure the reliability of the results, a minimum of three repeated measurements were 504 carried out, aiming to attain good reproducibility.”

Reviewer: Standard deviations should be presented.

ect.

Author Response

Manuscript Title: Exploring the Efficacy of Benzimidazolone Derivative as Corrosion Inhibitors for Copper in NaCl 3.5 wt.% Solution: A Comprehensive Experimental and Theoretical Investigation.

"Response to Reviewer X Comments"

The authors would like to thank the reviewers for the precise and thoughtful comments, as well as their constructive criticism. The responses to the individual comments of the reviewer are detailed below.

Response to Reviewer # 1:

Reviewer In the Introduction, the authors should focus on copper, not on mild steel. Please rewrite.

Response: Thank you for pointing this out. The introduction was modified.

Reviewer The authors used “a saturated calomel electrode (SCE)”, but on all the figures and the text appears ECS?  

Response: We have corrected the notation accordingly

Reviewer The Kads highly differ, so how (delta)Gads can be practically the same? Also sign for “kilo” is k, not K.Tables 3 and 6.

Response: Using Equation 8, we found approximately the same value because the Ln(Kads) values are close. Furthermore, the two techniques used provide complementary information to each other.

Reviewer: What is the meaning of such a high cathodic Tafel slope (some are positive?) For the diffusion-controlled oxygen reduction reaction Tafel slope is meaningless. How the authors determined corrosion current density???? As authors stated, row 129, “Polarization curves for copper in a corrosive aqueous NaCl solution (3.5 wt.%) with-129 out and with different doses of CHBI are shown in Fig. 3(a), and the electrochemical parameters adjusted by Tafel analysis are given in Table 3,.....” What is the Tafel analysis; it can be applied only to the anodic part of the curve.

Response: at line 162, we have described that an interesting observation is that the cathodic branches of the polarization curves remain parallel to the blank solution, suggesting minimal alterations in the cathodic reaction. That high value of cathodic slope indicate the diffusion phenomena (oxygen reduction reaction). For Tafel analysis, we have used the EC-Lab software to determine the electrochemical parameters.

Reviewer: When using log functions, the correct is to write log (i / mA/cm2) or log (Z / Ohm cm2) not log i (mA/cm2)

Response: Thank you very much for your Remarque. Done

Reviewer: Tables, 4 and Fig. 3 Units for Z and R are Ohm cm2 not Ohm/cm2

Response: The unity of R and Z are corrected.

row 504 “To ensure the reliability of the results, a minimum of three repeated measurements were 504 carried out, aiming to attain good reproducibility.”

Reviewer: Standard deviations should be presented.

Response: we have added the Standard deviations.

Reviewer 2 Report

The article focuses on the synthesis, theoretical analysis, and application of a corrosion inhibitor known as benzimidazolone, specifically 1-(cyclohex-1-enyl)-1,3-dihydro-2H-benzimidazol-2-one (CHBI). The article discusses the structure determination of CHBI using X-ray diffraction (XRD) and evaluates its inhibitory properties on copper corrosion in a 3.5 wt.% NaCl solution using various electrochemical techniques such as potentiodynamic polarization curves (PDP) and electrochemical impedance spectroscopy (EIS). The article also mentions the use of scanning electron microscopy with energy dispersive X-ray spectroscopy (SEM-EDX), UV-visible spectroscopy, and theoretical calculations to further analyze the inhibitory performance of CHBI. The results suggest that CHBI acts as an excellent inhibitor, exhibiting a high inhibition rate of 86.49% at 10^-3 M. Density functional theory (DFT) and Monte Carlo (MC) simulation studies were also conducted to confirm the adsorption of the inhibitory molecule on the copper surface. Overall, the article provides insights into the synthesis, characterization, and inhibitory performance of CHBI as a corrosion inhibitor for copper. The results are clear. However, some revisions are still recommended.

1.In the first line of abstract, change "is focuses" to "focuses".

2. In line 18, change "dihydro-2H-benzimid- 18" to "dihydro-2H-benzimidazol-2-one (CHBI)".

3.In line 29, change "inhibitory mol- 25" to "inhibitory molecule on".

4.Introduction should be extended. Please provide a clearer introduction that highlights the significance of corrosion prevention and the role of organic inhibitors in this field. Also, please expand on the existing literature on corrosion inhibitors and their effectiveness in various environments.

5. Provide more context and background information on benzimidazolone derivatives and their potential applications.

6. Explain the specific experimental techniques used to evaluate the inhibitory properties of CHBI in more detail, including their principles and advantages.

7. Discuss the theoretical calculations and simulations conducted to understand the adsorption of CHBI on the copper surface, providing a brief explanation of DFT and Monte Carlo simulation methods.

8. Emphasize the key findings of the study, such as the inhibition rate of CHBI and its strong adsorption on the copper surface.

9. The English of the article is far below the standard. The paper has lots of grammatical mistakes and ill-constructed sentences.

Minor editing of English language required

Author Response

Manuscript Title: Exploring the Efficacy of Benzimidazolone Derivative as Corrosion Inhibitors for Copper in NaCl 3.5 wt.% Solution: A Comprehensive Experimental and Theoretical Investigation.

"Response to Reviewer X Comments"

The authors would like to thank the reviewers for the precise and thoughtful comments, as well as their constructive criticism. The responses to the individual comments of the reviewer are detailed below.

Response to Reviewer # 2:

1.In the first line of abstract, change "is focuses" to "focuses".

Response: Abstract: This study focuses on the synthesis,…...

2. In line 18, change "dihydro-2H-benzimid- 18" to "dihydro-2H-benzimidazol-2-one (CHBI)".

Response: dihydro-2H-benzimidazol-2-one (CHBI).

3.In line 29, change "inhibitory mol- 25" to "inhibitory molecule on".

Response: inhibitory molecule on……

4.Introduction should be extended. Please provide a clearer introduction that highlights the significance of corrosion prevention and the role of organic inhibitors in this field. Also, please expand on the existing literature on corrosion inhibitors and their effectiveness in various environments.

Response:

5. Provide more context and background information on benzimidazolone derivatives and their potential applications.

Response: Benzimidazolone derivatives have a wide range of potential applications, including pharmaceuticals, inhibitors, herbicides, pigments and fine chemicals. In addition, benzimidazoles, they are also of interest in advanced materials science, such as nonlinear optics. etc.

6. Explain the specific experimental techniques used to evaluate the inhibitory properties of CHBI in more detail, including their principles and advantages.

Response: the techniques used for electrochemical analyses were added in the experimental parts (line 507: Electrochemical measurement). For more explication about the principles and advantages, we have added some recent references.

7. Discuss the theoretical calculations and simulations conducted to understand the adsorption of CHBI on the copper surface, providing a brief explanation of DFT and Monte Carlo simulation methods.

Response: in the most stable equilibrium configurations, the aromatic ring of CHBI was aligned parallel to the Cu surface. Moreover, the oxygen and nitrogen atoms were oriented towards the metal surface Fig. 9. This flat orientation is possibly due to the formation of coordination and back bonding between Cu metal and CHBI inhibitor. It is herein evident that un-occupied copper orbitals (3d) will prefer to accept electrons from the adsorbed CHBI. In addition, CHBI has a lone pair of electrons on the active centers (-NH, -N-, =O), as well as π-electrons in the benzene ring. These electrons cloud provide sufficient electronic charge to the vacant orbitals (3d) of Cu, forming stable coordination bonds (chemisorption). The spontaneous position of CHBI inhibitor allows maximum interactions between its active sites (the heteroatoms and the π electrons of the phenyl ring) and the metal surface. As a result, this increases its coverage and promotes the formation of a protective layer on the copper surface.

8. Emphasize the key findings of the study, such as the inhibition rate of CHBI and its strong adsorption on the copper surface.

Response: in the conclusion, we have discussed the findings results

9. The English of the article is far below the standard. The paper has lots of grammatical mistakes and ill-constructed sentences.

Comments on the Quality of English Language

Response: We have checked the English language of the paper.

Reviewer 3 Report

The authors of this article investigate the excellent anti-corrosion properties of benzimidazole derivatives against copper acting in NaCl 3.5% solution. There are still some problems in the article, and the author is recommended to improve it. It is suggested that this manuscript can be accepted after minor revisions. Listed below are detailed comments on the author's manuscript.

1.       The English language needs revision as there are lots of mistakes.

2.       The Introduction section should be elevated.

3.       The inhibition rate in the Abstract section (86.49%) is not the same as the inhibition rate in the Conclusion section (86%), so please check.

4.       Figure 9 is not clear and it is recommended that it be redrawn.

Author Response

Manuscript Title: Exploring the Efficacy of Benzimidazolone Derivative as Corrosion Inhibitors for Copper in NaCl 3.5 wt.% Solution: A Comprehensive Experimental and Theoretical Investigation.

"Response to Reviewer X Comments"

The authors would like to thank the reviewers for the precise and thoughtful comments, as well as their constructive criticism. The responses to the individual comments of the reviewer are detailed below.

Response to Reviewer # 3:

1.   The English language needs revision as there are lots of mistakes.

Response: We have checked the English language of the paper

2. The Introduction section should be elevated.

Response: The introduction was modified and rewrite.

3.  The inhibition rate in the Abstract section (86.49%) is not the same as the inhibition rate in the Conclusion section (86%), so please check.

Response: Thank you for your Remarque, the inhibition rate is 86.49% at 10-3

4.   Figure 9 is not clear and it is recommended that it be redrawn.

Response: Figure 9 is reworded in the text (Figure 10).

Reviewer 4 Report

The manuscript “Exploring the Efficacy of Benzimidazolone Derivative as Corrosion Inhibitors for Copper in NaCl 3.5 wt.% Solution: A Comprehensive Experimental and Theoretical Investigation” is devoted to the investigations of corrosion inhibitory activity of one heterocyclic compound by a standard set of physical and physicochemical methods. Although the research is quite routine, it has a pronounced practical nature and can be in demand in industry.

In my opinion, this manuscript suits to the scope of Molecules. I recommend that after monor revision, it can be accepted.

Some comments for the authors:

1. Lines 60-61, a figure with structures is required for “benzimidazole derivatives possess inhibitive properties”.

2. Line 97: for additional document, a reference is required.

3. For CHBI inhibitor, 1H NMR and 13C NMR spectra plots should be provided in supporting information.

4. Section 3.1 should be presented in a standard format for reports on organic compounds (for example, see https://www.mdpi.com/1420-3049/28/18/6636).

5. Possibly, a comparison of effectiveness (including economical) of the investigated compound with commercial inhibitors should be provided in the text.

This study is focuses on the synthesis,

Author Response

Manuscript Title: Exploring the Efficacy of Benzimidazolone Derivative as Corrosion Inhibitors for Copper in NaCl 3.5 wt.% Solution: A Comprehensive Experimental and Theoretical Investigation.

"Response to Reviewer X Comments"

The authors would like to thank the reviewers for the precise and thoughtful comments, as well as their constructive criticism. The responses to the individual comments of the reviewer are detailed below.

Response to Reviewer # 4:

1. Lines 60-61, a figure with structures is required for “benzimidazole derivatives possess inhibitive properties”.

Response: A figure showing the structures of benzimidazoles as corrosion inhibitors is added to the introduction.

2. Line 97: for additional document, a reference is required.

Comments: A reference has been added to the "chemical and synthesis procedure" section, line 105.

3. For CHBI inhibitor, 1H NMR and 13C NMR spectra plots should be provided in supporting information.

Response: NMR (1H and 13C) and HRMS spectra are given in the supplementary data.

4. Section 3.1 should be presented in a standard format for reports on organic compounds (for example, see https://www.mdpi.com/1420-3049/28/18/6636).

Response: Section 3.1 is presented in a standard format for reports on organic compounds.

5. Possibly, a comparison of effectiveness (including economical) of the investigated compound with commercial inhibitors should be provided in the text.

Response: Thank you for your Remarque, we have added this part in the introduction.

Round 2

Reviewer 1 Report

Dear authors,

Many thanks for accepting suggestions and improving the manuscript to the stage of acceptance.

Best wishes 

Reviewer 2 Report

The corrections are nice. The article can be accepted in the current version.